# Ultrasound-Assisted Extraction of Total Phenolic Compounds and Antioxidant Activity Evaluation from Oregano (*Origanum vulgare* ssp. *hirtum*) Using Response Surface Methodology and Identification of Specific Phenolic Compounds with HPLC-PDA and Q-TOF-MS/MS

**DOI:** 10.3390/molecules28052033

**Published:** 2023-02-21

**Authors:** Afroditi Michalaki, Haralabos C. Karantonis, Anastasia S. Kritikou, Nikolaos S. Thomaidis, Marilena E. Dasenaki

**Affiliations:** 1Laboratory of Food Chemistry, Biochemistry and Technology, Department of Food Science and Nutrition, School of the Environment, University of The Aegean, 81400 Lemnos, Greece; 2Laboratory of Analytical Chemistry, Department of Chemistry, National and Kapodistrian University of Athens, 15771 Athens, Greece; 3Laboratory of Food Chemistry, Department of Chemistry, National and Kapodistrian University of Athens, 15771 Athens, Greece

**Keywords:** oregano, box–behnken, phenolics, antioxidants, HPLC-PDA analysis, UPLC-Q-TOF MS

## Abstract

Oregano is native to the Mediterranean region and it has been reported to contain several phenolic compounds particularly flavonoids that have been related with multiple bioactivities towards certain diseases. Oregano is cultivated in the island of Lemnos where the climate promotes its growth and thus it could be further used in promoting local economy. The aim of the present study was to establish a methodology for the extraction of total phenolic content along with the antioxidant capacity of oregano by using response surface methodology. A Box–Behnken design was applied to optimize the extraction conditions with regard to the extraction time, temperature, and solvent mixture with the use of ultrasound-assisted extraction. For the optimized extracts, identification of the most abundant flavonoids (luteolin, kaempferol, and apigenin) was performed with an analytical HPLC-PDA and UPLC-Q-TOF MS methodology. The predicted optimal conditions of the statistical model were identified, and the predicted values confirmed. The linear factors evaluated, temperature, time, and ethanol concentration, all showed significant effect (*p* < 0.05), and the regression coefficient (R^2^) presented a good correlation between predicted and experimental data. Actual values under optimum conditions were 362.1 ± 1.8 and 108.6 ± 0.9 mg/g dry oregano with regard to total phenolic content and antioxidant activity based on 2,2-Diphenyl-1-picrylhydrazyl (DPPH) assay, respectively. Additionally, further antioxidant activities by 2,2′-azino-bis(3-ethylbenzothiazoline-6-sulfonic acid (ABTS) (115.2 ± 1.2 mg/g dry oregano), Ferric Reducing Antioxidant Power (FRAP) (13.7 ± 0.8 mg/g dry oregano), and Cupric Reducing Antioxidant Capacity (CUPRAC) (1.2 ± 0.2 mg/g dry oregano) assays were performed for the optimized extract. The extract acquired under the optimum conditions contain an adequate quantity of phenolic compounds that could be used in the production of functional foods by food enrichment procedure.

## 1. Introduction

Oregano (*Origanum vulgare* ssp. *hirtum*) is an aromatic herb that mainly comes from *Lamiaceae* and *Verbenaceae families,* which are the most world’s commercially traded culinary herbs that has been used long as a condiment and spice for food, salads, meat, etc. [1,2]. *Origanum vulgare L*. originally came from warm climates in western and southwestern Eurasia and the Mediterranean region. It is a perennial plant that has the characteristics of an herb, green, and leaflike, with round shaped leaves. It is split into four main groups: Turkish oregano (*Origanum onites),* Spanish oregano (*Coridohymus capitatu*), Greek oregano (*Origanum vulgare*), and Mexican oregano (*Lippia graveolens*). The most studied oregano species is the Greek oregano (*Origanum vulgare*), with many studies explaining the potential as antioxidant, antimicrobial, antifungal, anti-inflammatory, and skin defensive auxiliaries with all these capacities to be associated with its rich polyphenolic content [3]. Oregano has been traditionally used in folk medicine for the treatment of general infections, inflammation-related illnesses, asthma, indigestion, stomachache, bronchitis, coughs, diarrhea, menstrual disorders, and diabetes [4,5]. Previous studies on the chemical composition of oregano have revealed the presence of phenolic acids and flavonoids. These compounds can potentially prevent the oxidating stress [6,7]. High antioxidant capacity is an important factor for the delay or the prevention of several diseases, such as heart diseases, neurodegenerative diseases, cancer, and of the aging process [8].

Oregano is known to contain a high quantity of phenolic acids and especially flavonoids. Among others, apigenin, luteolin, and kaempferol have been studied for its presence in oregano extracts [9,10]. Despite the well-known antioxidant properties of apigenin, this molecule has been also reported for the potential benefits on the immune system, sleep, testosterone production, blood sugar levels, and several types of cancer [11]. Furthermore, luteolin and kaempferol have been studied for the anti-oxidative, anti-tumor, and anti-inflammatory properties, but also for their potential anticancer properties [12,13,14]. In addition, oregano and its extracts are used to treat various illnesses, such as Alzheimer’s and cardiovascular diseases, but also are employed to help people with thrombosis problems and strokes [15]

Considering the health benefits of specific oregano constituents, it is of high interest to evaluate and optimize a procedure that will properly extract the compounds of interest in order to further utilize the extracts. Taking into consideration that the extraction step is of highest importance to acquire the compounds of interest; there is growing interest in evaluating the proper extraction techniques and optimize the respective process parameters.

Several new extraction techniques have been developed in the frame of green extraction, such as microwave extraction (ME), supercritical fluid extraction (SFE), and ultrasonic-assisted extraction (UAE). Special interest has been given in ultrasound assisted extraction due to its positive impact on bioactive compounds extraction process, such as higher product yields, shorter extraction time, and lower costs in contrast with other extraction techniques [16,17].

*Origanum vulgare* ssp. *hirtum* is cultivated on the island of Lemnos and is a crop important for the economic viability of the island. The study and highlighting of important bioactivities, such as its antioxidant activity, could lead to its further use as a raw material for the development of new functional foods. The goal of promoting innovation and entrepreneurship in the context of the agri-food sector is served within this study performed. The exploitation of the local herbs, such as *Origanum vulgare* ssp. *hirtum*, which is abundant in the island of Lemnos, could lead to the production of new functional foods or the further development of already existing traditional products. Increased consumer demand for such foods will in turn lead to more sustainable conditions in the future for local communities in these areas. Additionally, the increased consumer demand for such products would assist local economies in these locations by helping them establish a more solid economic basis for the future. At the same time, research in this area will continue to contribute towards this direction.

Thus, the aim of the present study was to identify the optimum extraction conditions of Oregano (*Origanum vulgare* ssp. *hirtum*) cultivated in the Greek island of Lemnos with regard to total phenolic compounds and the antioxidant activities of the extracts by using a response surface methodology. Additionally, for the extracts acquired under the optimum conditions, the existence of apigenin, kaempferol, and luteolin by HPLC-PDA and HPLC-QTOF-MS methodology was evaluated.

## 2. Results and Discussion

### 2.1. Model Fitting

The optimization of the extraction procedure by ultrasound assisted extraction (UAE) of total phenolic compounds and antioxidant activity evaluation by DPPH assay from oregano was carried out by response surface methodology. More specifically, a Box–Behnken Design (BBD) was used to find out the combined effect between the factors of extraction temperature (40 °C, 60 °C, and 80 °C) (Χ_1_), extraction time (20 min, 30 min, and 40 min) (Χ_2_), and ethanol concentration (60%, 70%, and 80% *v*/*v*) (Χ_3_).

The experimental design matrix produced based on BBD consisted of 15 combinations, including three center points, resulting in a randomizing run order to reduce impact of variation on response values owing to the external factors. The results obtained are shown in Table 1. The experimental values varied from 25.0 mg to 250.0 mg gallic acid equivalent/g dry oregano regarding the total phenolic compounds and 22.2 mg to 95.6 mg Trolox equivalent equivalent/g dry oregano regarding the DPPH assay (Table 1).

The fitting of the full quadratic approximation of the BBD response surface model was estimated by the analysis of variance (ANOVA). The F-values and relevant *p*-values were used to examine the significance of each source of terms, that is, linear, two-factor interaction, quadratic, and the regression coefficients of the fitted models. Terms with *p*-value lower than 0.05 at the 95% confidence interval were identified as statistically significant.

A multiple regression analysis was employed to fit the response value and the experiment data. Model reduction was carried out, excluding a lower-order term that did not affect the model hierarchy to further refine full quadratic response surface model by removing the insignificant terms with a significance level greater than 5% (*p* > 0.05). To further refine full quadratic response surface model, due to the existence of non-significance effect of factors, transformation of the data, and exclusion of time∗temperature term was performed.

The results of the analysis of variance (ANOVA), after data transformation and model reduction, that was used to determine the degree to which the quadratic approximation of the BBD response surface reduced models fitted the data are presented in Table 2.

The ANOVA results (Table 2) suggested that the refined second-order models were statistically significant for TPC and DPPH, since the F-values of 80.68 and 17.30 have a zero % and 0.001 % chance, respectively to occur due to noise.

Table 2 also indicates the linear, quadratic, and interaction terms that are significant for the models concerning the two tests performed, total phenolic content, and antioxidant activity evaluation. Concerning the linear terms temperature (X_1_), time (X_2_), and ethanol concentration (X_3_), they all showed significant effect (*p* < 0.05). Temperature, time, and ethanol as solvent have been previously shown to be factors of interest when optimization of phenolic compound green extraction is studied [18,19].

In addition to the linear source, quadratic terms indicated statistically significant effects. Quadratic term of temperature (X_1_^2^), time (X_2_^2^), and ethanol concentration (X_3_^2^) indicated statistically significant effects on both models (*p* < 0.05).

Moreover, concerning the interaction coefficient that exert statistically significant effects on the models, the interaction of time and ethanol concentration (X_1_X_3_), and temperature and ethanol (X_2_X_3_), showed statistically significant effect on the model of total phenolic compounds (*p* < 0.05).

The regression coefficient (R^2^) was also evaluated, results are presented in Table 2, indicating a good correlation between predicted and experimental data.

Depending on the substance and the molecules extracted, variations in ethanol may alter solution polarity, which could be extremely important for extraction by impacting phenolic solubility. On the other hand, high ethanol concentrations can result in pectin dehydration and protein denaturation, which prevent phenolics from diffusing through the matrix of plant material into the solution. Additionally, the right amount of water in the solution may cause the dry matter of plants to swell, expanding the contact surface between the solvent and the solute, performing a positive effect on the extraction [20]. More specifically, the amount of 60% ethanol has been successfully studied for its capacity to optimum extract phenolics from oregano [21].

Furthermore, in order to increase the mass transfer rate, cavitation effect, but also the solubility of phenolic compounds, solvent extraction is often carried out at relatively high temperatures, but high temperatures may also cause phenolic deterioration [22]. However, flavonoids degradation at high temperatures (greater that 150 °C) have been reported in prolonged extraction time (greater than 201 min) [23]. In the study performed by Oreopoulou et al., lower quantities of total phenolics where obtained compared to the quantity obtained in the present study. The immersion of oregano to water–steam distillation for 6 h, is possible to lead to degradation of some phenolics. Increased temperature improves the extraction efficiency of phenolics because it increases their solubility and diffusivity, which in turn improves the mass transfer. However, heat may be able to lessen the severity of cavitation bubbles collapsing by reducing the variations in vapor pressure between the interior and exterior of the bubbles. When the temperature of the extracted material is raised, the surface tension decreases, reducing the sheer force of the popping bubbles.

Extracting at the lowest possible cost is largely dependent on how quickly the procedure can be completed. Typically, better extraction efficiency may be seen during the first time periods owing to the steep gradient solvent slope, which gradually diminishes with time. In addition, short extraction times are achieved as a result of cavitation, thermal, and physical phenomena induced at the extracted material’s surface [24].

The lack of fit was also not significant (*p* > 0.05) for the models of TPC and antioxidant activity evaluation implying that the models fit the data and each model may give accurate predictions.

Significant linear, quadratic, and interaction terms lead to the predictive equations (Equations (1) and (2)) as presented in Table 3. Positive and negative signs, of its coefficient in its equation of the quadratic models, indicate, respectively, positive or negative effect on the extraction efficiency of the studied phenolics.

### 2.2. Optimization of the Extraction Conditions

Response surface methodology was employed to evaluate the combined effect of the three factors to maximize the extraction of the total phenolic compounds and the antioxidant activity of the extracts as per the DPPH assay. The three-dimensional response surface plots that describe the interactive effect of the independent factors on the quantity of phenolics that were extracted by UAE are presented, for all three factors which were shown to be significant, in Figure 1, Figure 2, Figure 3, Figure 4, Figure 5 and Figure 6.

The response surface methodology holds an important role in the exploration of the optimum conditions of independent variables that can contribute to achieve a maximum response [25,26]. Response surface plots are useful for establishing the response values and operation conditions as required. Additionally, they can provide a method to visualize the results and help in processing the experimental levels of each variable and the type of interactions between [27].

The 3D response surface plots in Figure 3 and Figure 6 show that extraction both for total phenolic compounds and the evaluation of antioxidant activity are favored in high values of ethanol in solvent (X_3_), and high values of temperature (T; X_1_). On the other hand, negative effect seems to have for both models the long experiment duration (X_2_) and high values of temperature (T; X_1_) as Figure 1 and Figure 4 indicate.

Based on the experimental results of the total phenolic content and the antioxidant activity evaluation, the extracted amounts shown in Table 4 of the specific combination of the three factors lead to maximum extraction. A temperature of 80 °C, time of 40 min, and ethanol of 60% (*v*/*v*) results in a maximum extraction of 362.1 ± 1.8 mg GAE/g DM with regard to the total phenolic content and 108.6 ± 0.9 mg TE/g DM with regard to the DPPH assay.

The optimal conditions were calculated with the response optimizer of the Minitab^®^ statistical software, and the results are presented in Table 4.

### 2.3. Verification of the Models

The validity of the predictive model was confirmed comparing the predicted and the experimental values at optimal conditions. The values predicted by the model at optimal conditions were 363.0 mg gallic acid equivalent per g of oregano, and 108.5 mg Trolox equivalent per g of dry oregano for total phenolic compounds and antioxidant capacity evaluation with regard to the DPPH assay, respectively, and the actual experimental values were 362.1 ± 1.8 and 108.6 ± 0.9 mg/g dry oregano. No significant differences were found between the predicted and the actual values (*p* > 0.05) indicating high accuracy of response optimization.

The desirability value may define the ideal solution’s degree of precision. The closer the desirability value is to 1, the greater the optimization precision. Therefore, the model validation and response values are not substantially different from the predictions under ideal circumstances. 

### 2.4. Antioxidan Activity Evaluation and Total Phenolic Determination

The antioxidant activity evaluation by the DPPH, ABTS, FRAP, and CUPRAC assays, and total phenolic content (TPC) of oregano optimized extracts are reported in Table 5.

Both ABTS and DPPH tests measure the ability of compounds to scavenge free radicals. Small differences between the DPPH and ABTS values of the optimized extract indicated that the phenolic compounds contributing to the free radical scavenging activity were compounds with comparable hydrophilicity as the ABTS assay is applicable to both hydrophilic and lipophilic antioxidant systems, whereas the DPPH assay is only applicable to hydrophobic antioxidant systems [28]. On the other hand, the FRAP and CUPRAC tests assess the sample’s capacity to reduce using ferric and cupric ions, respectively.

The results are promising since the higher amount of total phenolic compounds were extracted compared to studies that have been performed in the past, taking into consideration the advantage of the results obtained in this study, which concerns the green extraction perspective that was employed [29,30].

Antioxidants are used in foods to delay or prevent the oxidation of molecules. Natural and synthetic antioxidants are both viable options. Some synthetic antioxidants, such butylated hydroxy-anisole (BHA) and butylated hydroxytoluene (BHT), have been banned due to their carcinogenicity. As a result, phenolic compounds and other naturally occurring antioxidants are receiving more attention for potential application in the food enrichment process. Therefore, it is of significant importance to create natural antioxidants from plant matrices for nutritional reasons, and to enhance the nutritional profile of the goods [31].

### 2.5. Determination and Identification of Phenolic Compounds in Oregano Samples by HPLC-PDA and UHPLC-QTOF-MS

For the optimized extracts, identification and determination of apigenin, luteolin, and kaempferol were performed in oregano ultrasound assisted extract obtained with ethanol/water 60/40 (*v*/*v*) as a solvent for 40 min extraction duration and temperature was set at 80 °C.

Determination of phenolic compounds performed with the validated analytical method of HPLC-PDA. Figure 7, Figure 8 and Figure 9 show the profile of the phenolic compounds under evaluation for the standard and sample solution at the 340 nm. Luteolin, apigenin, and kaempferol were quantitively determined at 1.30 ± 0.05, 1.43 ± 0.06, and 0.40 ± 0.03 mg/g dry oregano, respectively.

The oregano extracts also analyzed with the use of UHLPC-QTOF-MS for confirmatory purposes, and the respective mass spectrums are presented at Figure 10. Precursor ions were obtained, and the identification performed with validated methodology along with the software library for the three phenolic compounds.

Identification of apigenin, luteolin, and kaempferol was based on the accurate mass measurements of the molecular ion [M-H]^−^ along with the fragmentation patterns of each molecule, the UV–Vis data and the comparison with the commercial standards that were available for these specific phenolic compounds.

Phenolic compounds have attracted attention with respect to their applications in foods and pharmaceutical matrices, especially considering the bioactive potential of these molecules [32]. The health advantages of apigenin, kaempferol, and luteolin have been extensively researched [33,34]; therefore, it would be of great interest to enrich foods with extracts containing an adequate amount of these phenolics, and further assess their properties. Due to the use of human-friendly solvents, these extracts may be used in the process of food enrichment to improve their nutritional profile. However, there are some variables that must be considered before going with oregano extract food fortification. One critical characteristic is the stability of the extracts and the examination of approaches to increase stability, such as the encapsulation of the extracts’ components of interest [35].

## 3. Materials and Methods

### 3.1. Chemicals and Reagents

Organic oregano (*Origanum vulgare* ssp. *hirtum*), native in the Greek island of Lemnos (39°55′ N 25°15′ E) and harvested during May of 2022, used in the experiments was kindly offered by Aegean Organics Ltd. (Lemnos, Greece). Oregano leaves were used. The reagents Folin–Ciocalteau, Trolox (6-hydroxy-2,5,7,8-tetremethychroman-2-carboxylic acid) and anhydrous sodium carbonate were purchased from SDS (Peypin, France). DPPH (1, 1-Diphenyl-2-picryl-hydrazyl), caffeic acid, luteolin, and apigenin were purchased from Sigma–Aldrich (St. Louis, MO, USA). Methanol, acetic acid HPLC water, and acetonitrile were purchased from Thermo-Fisher scientific (Nepean, ON, Canada). Neocuproine was purchased from Acros Organics (Fair Lawn, NJ, USA). Ammonium acetate, sodium chloride, sodium dihydrogen phosphate dehydrate, and copper chloride dihydrate were all purchased from Penta (CZ Ltd., Chrudim, Czech Republic). ABTS (2,20 -Azino-bis-(3-ethylbezothiazoline-6-sulphonic acid was purchased from Applichem (Darmstadt, Germany). Potassium persulfate was purchased from Chem-Lab (Zedelgem, Belgium).

### 3.2. Preparation of the Samples

For the preparation of the oregano samples, the already dried oregano (moisture < 10%) was processed for one minute in a laboratory grinder, IKA A 10 basic (IKA Works, Wilmington, DE, USA), to get a sample of fine powder.

### 3.3. Ultrasound-Assisted Extraction (UAE)

The phenolic compounds extraction from oregano samples performed with the use of a threaded end of 1.000 mL maximum volume cup-horn of 750 Watt ultrasonic processor VCX-750 equipped with a sealed converter (Sonics & Materials, Inc. Newtown, CT, USA). A one-to-one pulse in seconds was applied combined with a 60% amplitude, while several temperatures were applied for 25 °C to 80 °C for the preliminary experiments. With regard to the sample preparation, 0.25 g of dried samples were weighed in a 10-mL tube. The tube was filled up to 5.0 mL with Purified water, ethanol, or ethanol/water in various ratios. After extraction, samples were centrifuged for 5 min at 3000× *g* and supernatants were filtered in HPLC vials through 0.20-μm RC (regenerated cellulose) filters before the analysis.

### 3.4. Evaluation of Antioxidant Activity

The antioxidant activities of the extracts, obtained at optimized conditions, were evaluated by the DPPH, ABTS, FRAP, and CUPRAC assays. Each sample was examined in triplicate. Trolox solutions were prepared in appropriate concentrations for quantitation purposes and the results expressed as Trolox equivalents in mg per g of oregano for all the antioxidant tests performed.

The capacity of extracts to scavenge the free radical of DPPH was evaluated by the method of Brand-Williams et al. [36] [NO_PRINTED_FORM] with minor modifications. An aliquot of the extract (2 to 10), or an appropriate standard solution of Trolox was diluted with methanol up to 0.9 mL. Then, 0.1 mL of 0.6 mM DPPH reagent in methanol was added, followed by vigorous stirring. After 15.0 min in the dark, the absorbance was measured at 515 nm against a reference sample containing methanol.

Determination of ABTS radical scavenging activity of samples was performed by the method of Miller et al. [37] with minor modifications. ABTS radical cation (ABTS^•+^) was produced by the oxidation of ABTS with potassium persulfate (K_2_S_2_O_8_). The ABTS^•+^ was generated by reacting 7 mmol/L stock solution of ABTS with potassium persulphate in a final concentration equal to 2.45 mmol/L. The ABTS^•+^ working solution was prepared by dilution of the stock solution using distilled water to give an absorbance of 0.700 at 734 nm. Aliquots of parsley extracts (2 to 10 µL), or appropriate amounts of Trolox standards were diluted to 1.0 mL with working ABTS^•+^ solution and were vigorously stirred. Samples remained for 15.0 min in the dark at ambient temperature and the absorbance was measured at 734 nm. The ability of the extracts to scavenge the ABTS^•+^ was evaluated relative to a reference sample that did not contain any quantity of extract.

The reducing potential of the samples was determined using the FRAP assay as described by Benzie and Strain [38]. The method is based on the reduction in the Fe^3+^—tripyridyl triazine complex to its ferrous-colored form at low pH in the presence of antioxidants. The FRAP reagent was freshly prepared and contained 0.2 mL of a 10 mM TPTZ (2,4,6-tripyridy-s-triazine) solution in 40 mM HCl plus 0.2 mL of 20 mM FeCl_3_•6H_2_O plus 0.2 mL of 3.0 M acetate buffer, pH 3.6. Aliquots of extracts (2 to 10 µL) were transferred in test tubes, and dissolved up of 900 µL with distilled water, followed by addition of 300 µL of FRAP solution and vigorous stirring. The samples were incubated for 10 min in a 37 °C water bath, and the absorbance was measured at 593 nm.

The reducing capacity of the samples was also determined using the CUPRAC assay according to Özyürek et al. [39]. Aliquots of parsley extracts (2 to 10 µL) were transferred in test tubes and diluted with 300 µL of 10 mM CuCl_2_•2H_2_O, 7.5 mM neocuproine, and 1 mM CH_3_COONH_4_ buffer solution with pH = 7.0, followed by the addition of distilled water up to the volume of 1200 µL. The samples were well stirred and remained at room temperature for 30 min. The absorbance of the samples was then measured at 450 nm.

### 3.5. Determination of Phenolic Compounds

The total content of phenolics in oregano extracts obtained at optimized conditions were measured in triplicate by using a modified version of Singleton and Rossi’s technique and was determined using the Folin–Ciocalteu’s method with some modifications [40] using a spectrophotometer Lambda 25 (Perkin Elmer, Norwalk, CT, USA). The experiment was carried out by combining 1 to 10 µL of oregano extracts with 1.8 mL of distilled water and 0.1 mL of Folin–Ciocalteu reagent. The materials were then rapidly mixed and incubated in the dark for two minutes. After adding 0.3 mL of 20% (*w*/*v*) aqueous Na_2_CO_3_, the samples were rapidly agitated and incubated at 40 °C in a water bath for 30 min. Absorbance was measured spectrophotometrically at 765 nm, by a Spectrophotometer Lambda 25 (Perkin Elmer, Norwalk, CA, USA). Gallic acid was used to develop a standard curve. The final findings were expressed as equivalent concentrations of gallic acid (mg GAE per g oregano).

### 3.6. Determination of Phenolic Compounds with HPLC-PDA

For the determination of phenolic compounds during preliminary experiments, but also during the analysis of the optimized extracts produced in the frame of experimental design, a Shimadzu HPLC 2030C prominence-i system was used, equipped with a binary pump, a degasser, an autosampler, a column heater, and a PDA detector. A Phenomenex Luna C18(2) analytical column (4.6 mm × 250 mm, particle size 5.0 μm) was used for the separation of the phenolic compounds under evaluation. The elution was performed using water acidified with 0.2% formic acid (mobile phase A) and methanol (mobile phase B). The adopted elution gradient was applied as follows: 0 min, 5% mobile phase B; 1 min, 5% mobile phase B; 30 min, 95% mobile phase B; 30.1 min, 5% mobile phase B; and 33 min, 5% mobile phase B. The injection volume was 20 μL, UV–vis spectra were recorded from 190 to 800 nm, while the chromatograms were registered at 280 and 340 nm. The analytical methodology was successfully validated in terms of linearity, accuracy, stability, limit of quantitation, and precision (system precision and reproducibility) for each phenolic compound under evaluation [41]. Quantification and identification performed with the use of commercial standards that were available for the phenolic compounds under evaluation.

### 3.7. Identification of Phenolic Compounds with HPLC-QTOF-MS

An UHPLC system with an HPG-3400 pump (Dionex UltiMate 3000 RSLC, Thermo Fisher Scientific, Germany) coupled to a QTOF mass spectrometer (Maxis Impact, Bruker Daltonics, Bremen, Germany) was used for the analysis. Negative electrospray ionization mode was applied. Separation was carried out using an Acclaim RSLC C18 column (2.1 × 100 mm, 2.2 μm) purchased from Thermo Fisher Scientific (Driesch, Germany) with a pre-column of ACQUITY UPLC BEH C18 (1.7 μm, VanGuard Pre-Column, Waters (Waters Corporation^®^, Wexford, Ireland). Column temperature was set at 30 °C. The solvents used consisted of (A) 90% H_2_O, 10% methanol, and 5 mM CH_3_COONH_4_ (Mobile phase A), and 100% methanol and 5 mM CH_3_COONH_4_ (Mobile phase B). The adopted elution gradient started with 1% of organic phase B with flow rate 0.2 mL min−1 during 1 min, gradually increasing to 39% for the next 2 min, then increasing to 99.9% and flow rate 0.4 mL min^−1^ for the following 11 min. These almost pure organic conditions were kept constant for 2 min (flow rate 0.48 mL min^−1^), then initial conditions (1% B–99% A) were restored within 0.1 min (flow rate decreased to 0.2 mL min^−1^) to re-equilibrate the column for the next injection.

The QTOF-MS system was equipped with an electrospray ionization interface (ESI), operating in negative mode with the following settings: capillary voltage of 3500 V, end plate off-set of 500 V, nebulizer pressure of 2 bar (N_2_), drying gas of 8 L min^−1^ (N_2_), and drying temperature of 200 °C. A QTOF external calibration was daily performed with sodium formate (cluster solution), and a segment (0.1–0.25 min) in every chromatogram was used for internal calibration, using calibrant injection at the beginning of each run. The sodium formate calibration mixture consisted of 10 mM sodium formate in a mixture of H_2_O/isopropanol (1:1). Full scan mass spectra were recorded over the range of 50–1000 m/z, with a scan rate of 2 Hz. MS/MS experiments were conducted using AutoMS data-dependent acquisition mode based on the fragmentation of the five most abundant precursor ions per scan. The instrument provided a typical resolving power (FWHM) between 36,000 and 40,000 at m/z 226.1593, 430.9137, and 702.8636, respectively. Identification was performed with the use of commercial standards that were available for the phenolic compounds under evaluation along with the fragmentation patterns of each molecule.

### 3.8. Experimental Design

The Box–Behnken design (BBD), a standard RSM design, is highly suited to fitting a quadratic surface, which is often used for process optimization, was selected to identify the optimum extraction conditions for total phenolic compounds and antioxidant activity measurements. The three independent factors were temperature (X_1_), time (X_2_), and ethanol concentration (X_3_).

Each factor was coded ta three levels: −1, 0, and +1. One replicate experiment was performed. The factors and their corresponding levels, both coded and actual, chosen in the three-factor-three-level BBD were based on preliminary one-factor-at-a-time experiments, literature research, and instrumental specifications, and are presented in Table 6.

RSM was used to fit a complete second-order polynomial equation to the design points and experiment data. The following quadratic response surface model equation (a) for four components was fitted:(3)$$Y=\beta_{0}+\overset{3}{\underset{i=1}{\sum }}\beta_{i}X_{i}+\overset{3}{\underset{i}{\sum }}\beta_{ii}X_{i}^{2}+\overset{2}{\underset{i=1}{\sum }}\overset{3}{\underset{j=i+1}{\sum }}\beta_{ij}XiXj+\varepsilon$$

In Equation (3), *Y* corresponds to the response variable expressed by mg GAE/g and mg TE/g for TPC and DPPH, respectively, for both tests performed (total phenolic content and antioxidant evaluation). *Xi* and *Xj* are the independent factors affecting the response (Table 1). The terms *β*_0_, *β_i_*, *β_ii_*, and *β_ij_* are the regression coefficients of the model (intercept, linear, quadratic, and interaction term), and *ε* corresponds to the random error term.

Analysis of variance (ANOVA) was used to estimate the fitting of the entire quadratic approximation of the BBD response surface model. The significance of each source of terms (linear, two-factor interaction, and quadratic), and the regression coefficients of the fitted model were examined using the F-values and pertinent *p*-values. Statistically significant terms were those whose probability (*p*-value) at the 95% confidence level fell below 0.05.

### 3.9. Verification of the Statistical Model

Optimum extraction conditions of the total phenolic content and antioxidant activity evaluation of oregano samples based on the evaluation for extraction temperature, and time and solvent composition were obtained using the predictive equations of RSM. The obtained concentration was determined after extraction of phenolic compounds under optimal conditions. The experimental and predicted values were compared to determine the validity of the model.

### 3.10. Statistical Analysis

Data presented as mean ± standard deviation (m ± SD) for triplicate measurements. The response values of the RSM model for one replication with three center points were analyzed by Minitab^®^ trial version statistical software (Minitab Ltd., Coventry, UK). SPSS V 28.0.10 software (IBM Corp., Armonk, NY, USA) was used for one-sample t-test analysis for the verification of the model. Statistical significance was defined at <0.05.

## 4. Conclusions

In this study, a Box–Behnken Box–Behnken design (BBD), along with Box-Cox transformation of the data and model reduction, have been developed to optimize the extraction conditions for maximum total phenolic extractions and antioxidant activity based on DPPH assay from Oregano (*Origanum vulgare* ssp. *hirtum*) cultivated in Lemnos.

For the optimized extracts, the antioxidant activity was also measured with ABTS, FRAP, and CUPRAC assays. Then, luteolin, kaempferol, and apigenin were determined by HPLC-DAD and identified by UHPLC-Q-TOF-MS in extracts obtained by UAE from oregano sample.

The adequacy of the predictive model and the verification of the model were confirmed. Optimal conditions were calculated and found the same both for total phenolic contain, and antioxidant activity evaluation (DPPH assay). Using a concentration of ethanol equal to 60% (*v*/*v*), a temperature of 80 °C, and a time period of 40 min, the predictive and the actual values along with the desirability were equal to 363.0, 362.1 ± 1.8, and 1.000 for TPC; and 108.5, 108.6 ± 0.9, and 1.000 for DPPH.

High values of total phenolic compounds equal to 362.1 ± 1.8 mg GAE per g of oregano were determined in the optimized extract, and high antioxidant capacities that ranged from 1.2 mg to 115.2 mg Trolox equivalent per g of oregano with respect to DPPH, ABTS, FRAP, and CUPRAC assays, respectively, were obtained (Table 5). Additionally, luteolin, apigenin, and kaempferol were quantitively determined at 1.30 ± 0.05, 1.43 ± 0.06, and 0.40 ± 0.03 mg/g dry oregano, respectively.

With these findings, an adequate amount of the phenolic compounds under consideration were extracted from Oregano, specifically apigenin, luteolin, and kaempferol (1.30 0.05, 1.43 0.06, and 0.40 0.03 mg/g dry oregano, respectively) using ultrasound-assisted extraction and green solvents, such as ethanol/water mixture. It is important to highlight the fact that the results are obtained under the green extraction rules, which means that the extracts can further be used in food enrichment procedures.

However, there are several factors that must be considered before the commercial implementation of the recovery of value-added chemicals from natural products. The bioavailability, but also the metabolism of phenolic compounds are important parameters that is necessary to be investigated in order to better understand the biological mechanisms, which will empower the development of better applications for the phenolic compounds. The considerable total phenolic content of oregano extracts presents an opportunity for the creation of novel functional foods or the refinement of current traditional products with superiority in consumer health protection. The results indicate that oregano extract could be the subject of a mixture design for the formulation of new enriched healthy animal or plant food products, such as meat products, dairy products, bakery snacks, traditional pasta, spread products, beverages, etc. The result of the study highlights the nutraceutical potential of extracts form oregano (*Origanum vulgare* ssp. *hirtum*) cultivated in Lemnos. Oregano is a widespread cultivation aromatic herb, and considerable amount is easily accessible for the creation of novel functional foods or the refinement of current traditional products with shown superiority in consumer health protection.

## Figures and Tables

**Figure 1 molecules-28-02033-f001:**
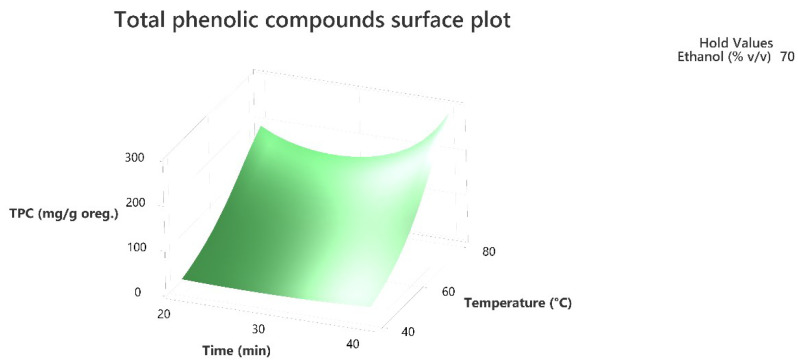
Three-dimensional response surface plot of the TPC (response variable) in mg of gallic acid equivalent/g of dry matter (DM) of oregano as a function of Time (X_2_) in min. and Temperature (X_1_) in Celsius degrees (°C) while holding ethanol at 70% (*v*/*v*).

**Figure 2 molecules-28-02033-f002:**
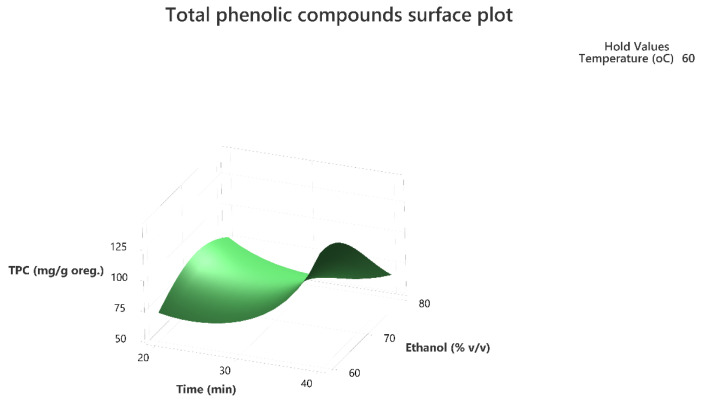
Three-dimensional response surface plot of the TPC (response variable) in mg of gallic acid equivalent/g of dry matter (DM) of oregano as a function of Time (X_2_) in min. and Ethanol %*v*/*v* (X_3_) while holding temperature at 60 °C.

**Figure 3 molecules-28-02033-f003:**
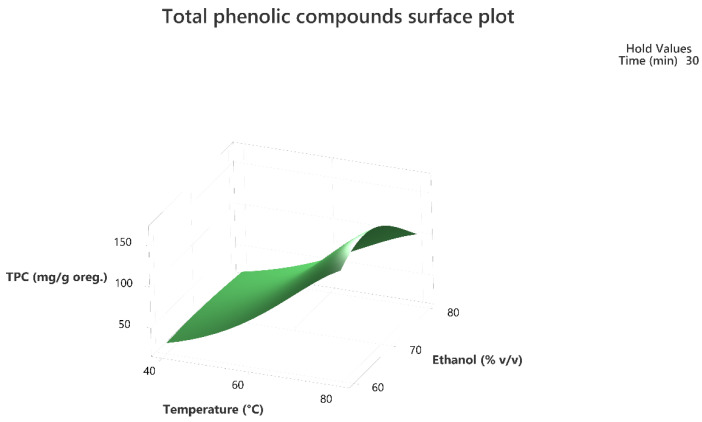
Three-dimensional response surface plot of the TPC (response variable) in mg of gallic acid equivalent/g of dry matter (DM) of oregano as a function of temperature (X_2_) in °C and Ethanol %*v*/*v* (X_3_) while holding time at 30 min.

**Figure 4 molecules-28-02033-f004:**
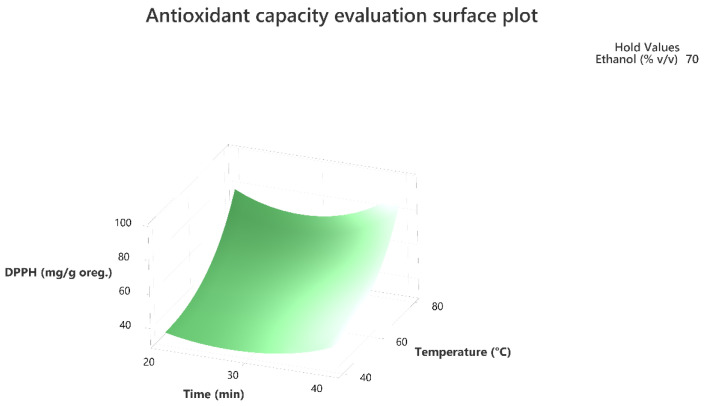
Three-dimensional response surface plot of the antioxidant capacity evaluation-DPPH assay (response variable) in mg of Trolox equivalent/g of dry matter (DM) oregano as a function of Time (X_2_) in min. and Temperature (X_1_) in Celsius degrees (°C) while holding ethanol at 70% (*v*/*v*).

**Figure 5 molecules-28-02033-f005:**
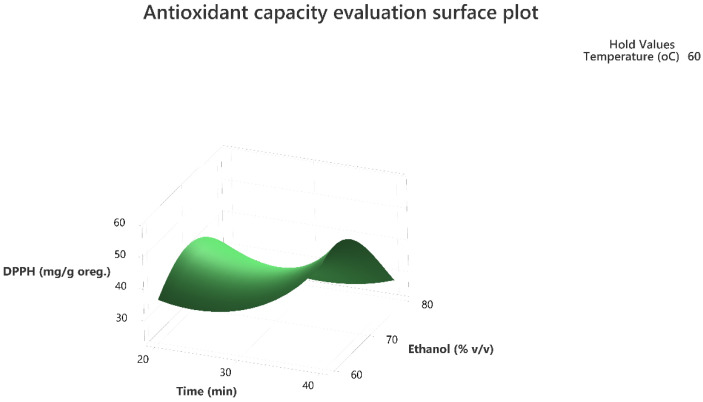
Three-dimensional response surface plot of the antioxidant capacity evaluation-DPPH assay (response variable) in mg of Trolox equivalent/g of dry matter (DM) of oregano as a function of Time (X_2_) in min. and Ethanol %*v*/*v* (X_3_) while holding temperature at 60 °C.

**Figure 6 molecules-28-02033-f006:**
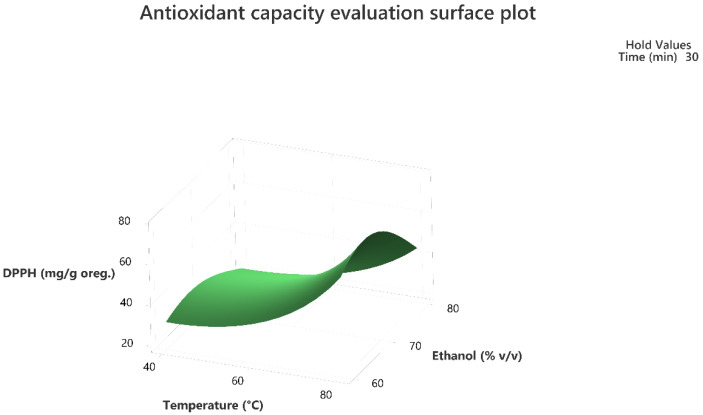
Three-dimensional response surface plot of the antioxidant capacity evaluation-DPPH assay (response variable) in mg of Trolox equivalent/g of dry matter (DM) of oregano as a function of temperature (X_2_) in Celsius degrees (°C) and Ethanol %*v*/*v* (X_3_) while holding time at 30 min.

**Figure 7 molecules-28-02033-f007:**
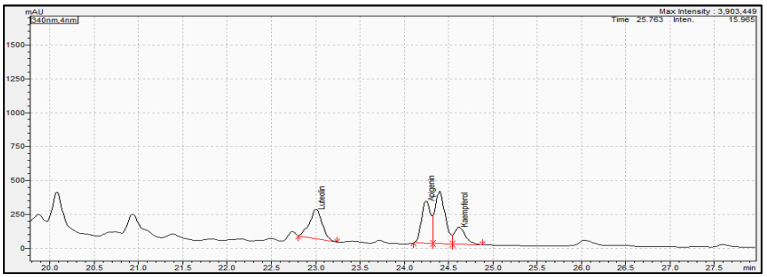
Representative chromatogram acquired for oregano sample extract at 340 nm. Red line in figure shows the integration area for the quantification of each compound.

**Figure 8 molecules-28-02033-f008:**
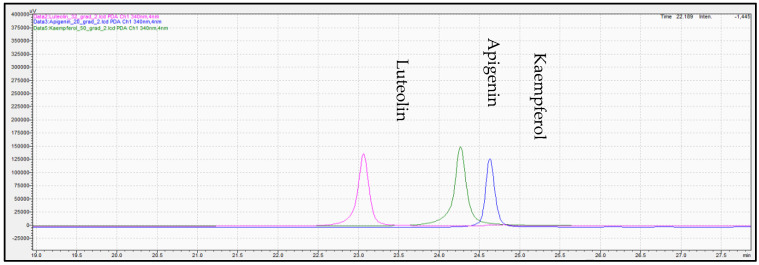
Overlay representative chromatogram acquired for standard solution of (pink color), apigenin (green color), and kaempferol (blue color) at 340 mn.

**Figure 9 molecules-28-02033-f009:**
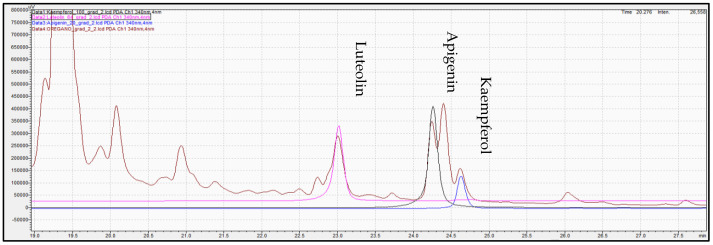
Overlay chromatogram acquired for standard solution of luteolin (pink color), apigenin (black color), kaempferol (blue color), and oregano extract (brown color) at 340 mn.

**Figure 10 molecules-28-02033-f010:**
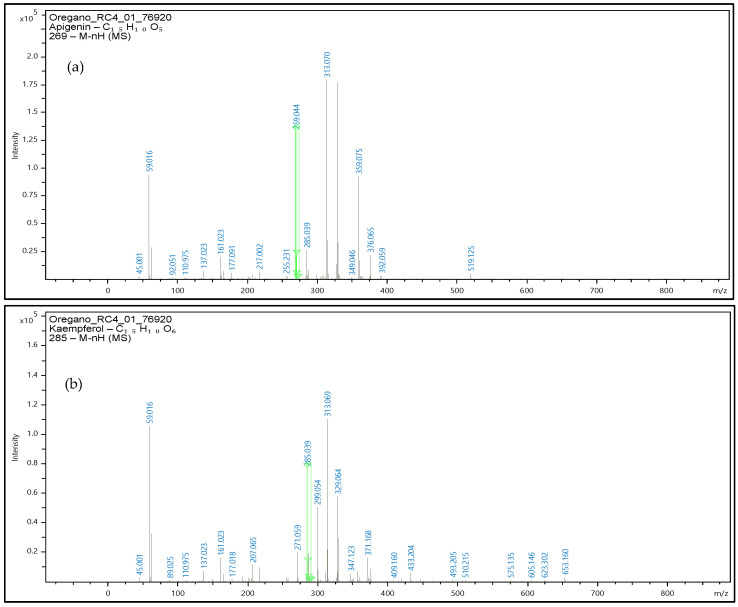
Mass spectrum of oregano extract samples at negative ionization were the precursor ion of apigenin- m/z 269 (**a**), kaempferol- m/z 285 (**b**), and luteolin- m/z 285 (**c**) are obtained. Green line in figure denotes the m/z value for each precursor ion.

**Table 1 molecules-28-02033-t001:** Coded and actual values of BBD design, and the results of experiments for total phenolic compounds and antioxidant activity evaluation by DPPH assay from oregano.

Run	Independent Factors			Dependndent Factors			
				Experimental Values		Predicted Values	
	X_1_ Temperature (°C)	X_2_ Time(min)	X_3_ Ethanol (% *v*/*v*)	TPC (mg/g) ^1^	DPPH (mg/g) ^1^	TPC (mg/g) ^1^	DPPH (mg/g) ^1^
1	60 (0)	20 (−1)	80 (+1)	80.0	30.4	77.2	30.8
2	40 (−1)	30 (0)	80 (+1)	25.0	22.2	23.8	20.3
3	80 (+1)	30 (0)	80 (+1)	95.0	41.3	103.6	43.9
4	60 (0)	30 (0)	70 (0)	73.3	46.8	78.1	38.8
5	80 (+1)	20 (−1)	70 (0)	180.0	71.7	194.4	78.4
6	40 (−1)	20 (−1)	70 (0)	30.9	39.3	31.5	36.2
7	60 (0)	40 (+1)	80 (+1)	64.7	27.7	67.4	28.2
8	40 (−1)	30 (0)	60 (−1)	25.0	26.2	28.3	30.6
9	80 (+1)	30 (0)	60 (−1)	205.0	75.2	150,7	66.4
10	60 (0)	40 (+1)	60 (−1)	134.8	55.9	141.4	55.2
11	40 (−1)	40 (+1)	70 (0)	40.9	42.2	36.6	43.0
12	60 (0)	30 (0)	70 (0)	69.5	31.6	78.1	38.8
13	80 (+1)	40 (+1)	70 (0)	250.0	95.6	289.3	93.2
14	60 (0)	20 (−1)	60 (−1)	75.0	36.5	71.9	35.8
15	60 (0)	30 (0)	70 (0)	95.0	39.6	78.1	38.8

^1^ DPPH (2,2-diphenyl-1-picrylhydrazyl): Results are presented as mg of Trolox equivalents (TE) per g of dry oregano; TPC: Total phenolic content presented as mg of Gallic acid equivalents per g of dry oregano.

**Table 2 molecules-28-02033-t002:** Results of the analysis of the variance (ANOVA) for transformed data concerning the fitting of reduced response surface model for UAE extraction of total phenolic content (TPC) and antioxidant activity evaluation (DPPH assay).

	^1^ TPC			^1^ DPPH		
Source	^2^ DF	F-Value	*p*-Value	DF	F-Value	*p*-Value
Model	8	80.68	0.000	8	17.30	0.001
Linear	3	194.68	0.000	3	31.87	0.000
Time	1	10.83	0.017	1	3.55	0.108
Temperature	1	550.09	0.000	1	71.62	0.000
EtOH Conc	1	23.13	0.003	1	20.45	0.004
Square	3	11.30	0.007	3	11.98	0.006
Time∗Time	1	17.93	0.005	1	8.37	0.028
Temperature∗Temperature	1	5.74	0.054	1	9.58	0.021
EtOH Conc∗EtOH Conc	1	7.68	0.032	1	15.50	0.008
2-Way Interaction	2	13.74	0.006	2	3.43	0.102
Time∗Temperature	1	-	-	1	-	-
Time∗EtOH Conc	1	14.24	0.009	1	4.05	0.091
Temperature∗EtOH Conc	6	13.24	0.011	6	2.81	0.145
Error	4			4		
Lack-of-Fit	2	0.10	0.973	2	0.14	0.953
Pure Error	14			14		
R^2^		0.9772		0.9390		
Adjusted R^2^		0.9544		0.8780		
Predicted R^2^		0.9108		0.8084		

^1^: Box-Cox data transformation was performed using optimal λ = −0.12 for TPC, and λ = 0.23 for DPPH. Reduction in the models was performed with the exclusion of temperature*time term without affecting the hierarchy model. ^2^: DF stands for Degree of freedom.

**Table 3 molecules-28-02033-t003:** Quadratic models of polynomial predictive equations of response surface total phenolic content (TPC) and antioxidant activity evaluation (DPPH assay) from oregano.

	^1^ Predictive Equations	
TPC	(Equation (1))	−TPC^−0.5^ −1.013 – 0.00053 X_2_ + 0.00852 X_1_ + 0.01692 X_3_ + 0.000142 X_2_^2^ – 0.000049 X_1_^2^ – 0.000104 X_3_^2^ – 0.000105 X_2_ X_3_
DPPH	(Equation (2))	ln(DPPH)= −8.39 – 0.0171X_2_ – 0.0431X_1_ + 0.389X_3_ + 0.001946 X_2_^2^ + 0.000520 X_1_^2^ – 0.002647 X_3_^2^ – 0.001301 X_2_X_3_

^1^ DPPH: Results are presented as mg of Trolox equivalents (TE) per g of dry oregano; TPC: Total phenolic content presented as mg of Gallic acid equivalents per g of dry oregano. X_1_: Temperature (T) in °C; X_2_: Time in min, X_3_: Ethanol concentration (%, *v*/*v*).

**Table 4 molecules-28-02033-t004:** Solution for maximum extraction of total phenolic content (TPC) and antioxidant activity evaluation (DPPH assay) from oregano.

Independent Factors ^1^	Predicted Values ^1^	Experimental Values	Desirability ^2^
TPC (mg/gDM)	363.0 ^a^	362.1 ± 1.8 ^a^	1.0000
DPPH (mg/gDM)	108.5 ^a^	108.6 ± 0.9 ^a^	1.0000

^1^: Independent factors were set at 80 °C (X_1_), 40 min DM (X_2_), and 60% (*v*/*v*) (X_3_) both for TPC and antioxidant activity evaluation (DPPH assay). Same letters in rows denote values of not statistically significant difference. ^2^ Both individual and composite desirability for maximum total phenolic content and antioxidant activity based on DPPH assay.

**Table 5 molecules-28-02033-t005:** Antioxidant activities of oregano optimized extract.

^1^ Parameters	Oregano Optimized Extract	Calibration Curve
DPPH (mg TE/g)	108.6 ± 0.9	y = −0.0849x + 0.625
ABTS (mg TE/g)	115.2 ± 1.2	y = −97.31x + 67.084
FRAP (mg TE/g)	13.7 ± 0.8	y = 58.018x − 2.944
CUPRAC (mg TE/g)	1.2 ± 0.2	y = 161.7x+ 0.7858
TPC (mg GAE/g)	362.1 ± 1.8	y = 0.018x + 0.102

^1^ DPPH, ABTS, FRAP, and CUPRAC: Results are presented as mg of Trolox equivalents (TE) per g of dry oregano; TPC: Total phenolic content presented as mg of Gallic acid equivalents per g of dry oregano. Results are expressed as mean ± SD in final reported results between the three replicates of the optimized extracts acquired. For the calibration curves presented, y is referred to the concentration of Trolox or gallic acid obtained, and x is referred to the absorbance acquired during the experimental procedure of each assay.

**Table 6 molecules-28-02033-t006:** Independent factors and their levels in the Box–Behnken Experimental Design.

		Factor Levels and Range		
Factors	Codes	−1	0	1
Temperature (°C)	X_1_	40	60	80
Time (min)	X_2_	20	30	40
Ethanol (%, *v*/*v*)	X_3_	60	70	80

## Data Availability

Not applicable.

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
