# Peer review of "Ultrasound-Assisted Extraction of Total Phenolic Compounds and Antioxidant Activity Evaluation from Oregano (Origanum vulgare ssp. hirtum) Using Response Surface Methodology and Identification of Specific Phenolic Compounds with HPLC-PDA and Q-TOF-MS/MS"

_molecules, 2023, doi:10.3390/molecules28052033_

Round 1

Reviewer 1 Report

Recommendation: Major

The manuscript Ultrasound-assisted extraction of total phenolic compounds and antioxidant activity evaluation from Oregano (Origanum vulgare ssp. hirtum) using response surface methodology and HPLC-PDA and Q-TOF-MS/MS identification of specific phenolic compounds, the methodology was reasonable and technically sound. But in general, there are significant problems.

Comments to the Author:

.

Below are some important suggestions.

Point 1. The title of the research is too long, I think it should be shortened.

Point 2 Write the DPPH expansion under Table 1.

Point 3. Instead of this sentence (Also, further antioxidant activities by ABTS, FRAP, and CUPRAC assays were performed for the extracts), express your results with numbers in the abstract section.

Point 4. Abbreviations used in the abstract should be clearly written first.

Point 5. In the introduction, write the Latin of the Oregano varieties in italics.

Point 6. Which probe was used in the ultrasonic processor.

Point 7. Specify the spectrophotometer used for the measurements

Point 8. Put the statistical part at the end in the material method section

Point 9. In the statistical section, indicate how many replications the analyzes were made.

Point 10. For Table 1, show the experimental results of the samples and the RSM estimation results in the same table. Give the standard deviations of the experimental results

Point 11. Correct the arguments expression in Table 1. Separate into dependent and independent variables at the top of the table

Point 12. The illustrations of the 3D graphics are too poor for magazine quality. My suggestion is that it should be drawn through a better program and the DPI quality should be increased. At the same time, instead of giving six figures, they can be combined and given together as a maximum of two graphics.

Point 13. In what sense was Desirability used in Table 4?

Point 14. In the applied variables, the highest TPC value was determined as 250 mg/g and DPPH as 95.6 mg/g. However, it is interesting that higher TPC and DPPH values are found in Table 4 under optimization conditions. Is there an error here? It is worth reviewing again.

Point 15. In Table 5, the units should be corrected to mg TE/g. For TPC, it should be corrected as mg GAE/g.

Point 16 “determination and identification of phenolic compounds in parsley samples by HPLC-PDA 296 and UHPLC-QTOF-MS”  parsley???

Point 17 Phenolic compounds should be discussed in detail with current literature.

Point 18 Avoid using bibliography in the conclusion part of the article. Summarize in your own words.

Reviewer 2 Report

In this article, the authors establish an interesting study for optimizing the ultrasonic extraction of phenolics and their antioxidant activity by response surface methodology as well as identifying the chemical profile of Origanum vulgare ssp. Hirtum by HPLC-PDA and Q-TOF-MS/MS.

However, the manuscript needs improvement to make it suitable for the Molecules journal.

My comments are below:

Comment 1

Line 39: The names of oregano family must be in italic.

Comment 2

Line 41: Oregano's scientific name is origanum vulgar, not oregano vulgare.

Comment 3

Line 45-47: Scientific names should be in italic.

Comment 4

The problematic of the topic’s choice and the aim of the study should be more clarified.

Comment 5

The authors are asked to give more information on the harvesting of the samples, geographical coordinates, harvesting period, and the part of the plant used.

Comment 6

More information about materials and equipment in the section “Ultrasound-assisted extraction (UAE)” must be given (Marque, company and country).

Comment 7

Line 384: Please correct the quadratic equation as follow (There is only 3 factors and  not 4):

Comment 8

please number the equations as 1; 2; 3...

Comment 9

Why did the authors optimize a single response about the antioxidant activity (DPPH). Simultaneous optimization of this response with the other techniques, ABTS, FRAP and CUPRAC would have brought great added value.

Comment 10

There is confusion between the references cited for the methods of determining antioxidant activity. Normally Muller et al. is for activity by ABTS and Brand-Williams is for activity by DPPH. Please correct.

Comment 11

Please add the equations for calculating % inhibition for the methods of determining antioxidant activity.

 Please also check the units of expression of the antioxidant activity. For example for DPPH, if it is IC50 it should be in mg/ml. If not mg Trolox equivalent/g.

Comment 12

Line 386; Please correct the sentence:” In equation (c) Υ corresponds to the response variable (mg/g DM)) to “In equation (c) Υ corresponds to the response variable expressed by mg GAE/g and mg TE/g for TPC and DPPH, respectively).

Comment 13

Line 448  Delete the sentence according to Singleton, Orthofer, and Lamuela-Raventos it is already mentioned.

Comment 13

Please give more information about how compounds were quantified (is the commercial standars were used??)

Comment 14

In table 1, results must be given as mean ± standard deviation. This will allow us to do an analysis of variance to compare the mean obtained for each trial. And then, to highlight the effect of changing procession parameters on the two studied responses.

Comment 15

In the section dealing with the statistical validation of the two models, the authors discussed the overall effect of the regression and the effects of the factors and their interactions. However, they did not provide information on the coefficient of determination R² and R² adjusted, which are essential to confirm the validity of the models. Please add this information.

Comment 16

In table 2, Why the interaction Time*Temperature was removed? The effect of this interaction term must appear in the table and then it must removed in the models if it was not statistically significant

Comment 17

Line 140; What do the authors mean by the values λ= -0.12 for TPC, and λ= 0.23??

Comment 18

I think there is a problem with the processing done by the MINITAB software. By re-analyzing the chosen box-Behnken design using three different software MINITAB, JMP and Design Expert, the results I obtained are different from those given by the authors.

Please verify.

Comment 19

The quality of the response surface plots are very poor, it is necessary to improve.

Reviewer 3 Report

The authors of the work titled “Ultrasound-assisted extraction of total phenolic compounds and antioxidant activity evaluation from Oregano (Origanum vulgare ssp. hirtum) using response surface methodology and HPLC-PDA and Q-TOF-MS/MS identification of specific phenolic compounds” used ultrasound technology to extract total phenolic compounds from Oregano to enhance the extraction efficiency while maintain the antioxidant activity. HPLC-PDA and UPLC-QTOF-MS/MS were used to identify the phenolic compounds from Oregano, and the antioxidant potential of extracted compounds were evaluated. Oregano has many bioactivities due to its rich polyphenolic content, which provides its commercial application in food and additives industry. After careful evaluation, I recommend the manuscript for publication in Molecules following a major revision of the appended comments. 

1.     The abstract should be improve. The sentences from line 26 to line 28 can be combined into one sentence. The sentence in line 29-30 should improve, which should put some idiographic data, such as: R2, and p value from the ANOVA analysis.

2.     The species of Oregano should be mentioned in the beginning of introduction instead of presenting in line 80.

3.     Line 49 should have punctuation.

4.     More information should be provided in Table 2, such as the R2, adjusted R2, predicted R2, C.V.%.

5.     The response surface images are too vague, which need to be improved.

6.     Since the authors didn’t present details to analyse the ionic fragments about each phenolic compounds, I suggest to use the standard sample to verify and identify. Or the authors should analyse the ionic fragments obtained from each compounds.

7.     Antigoni Oreopoulou (https://doi.org/10.1016/j.fbp.2020.07.017) reported that the polyphenol content in Oregano was 60 mg GAE/g dry, while the polyphenol content (362.1 ± 1.8 mg GAE/g DM) tested in this manuscript were significantly higher than the previous studies. Could the authors explain the reasons why the content has such significant difference.

8.      The conclusions need to improve. It is necessary to highlight the key conclusions of your study rather than discussion.

9.     “Origanum vulgare ssp. Hirtum” in line 552 should be italics, please check the manuscript again.

Round 2

Reviewer 1 Report

The authors made the necessary corrections in the article.

Author Response

We thank the reviewer for his final comment

Reviewer 2 Report

The problematic of the topic’s choice need more clarification

The two references Miller et al. and Brand-Williams et al. have been corrected in the reference list but are still reversed in the text.

In comment 11, we ask for the equation for calculating the percentage inhibition and not the equations of the calibration curves obtained for Trolox by the five antioxidant activity determination techniques 

The processing done with the MINITAB software is not well done and the numbers that appear in the analysis of variance table are not correct .
The experiments were not done in trplicates, so we can't trust their quality and the authors can't make comparison tests between the results of the experiments
The mathematical models (Tables 3) are not correct.

Author Response

Reviewer 2:

The problematic of the topic’s choice need more clarification

Response: More clarifications were incorporated (Lines 87-95 in the revised manuscript)

The two references Miller et al. and Brand-Williams et al. have been corrected in the reference list but are still reversed in the text.

Response: References were corrected also in the text.

In comment 11, we ask for the equation for calculating the percentage inhibition and not the equations of the calibration curves obtained for Trolox by the five antioxidant activity determination techniques 

Response: Kindly be informed that the results are presented in Trolox/gallic acid equivalent per gram of dry oregano and not as the percentage inhibition.

The processing done with the MINITAB software is not well done and the numbers that appear in the analysis of variance table are not correct.

Response: The results were updated and presented step by step, both before and after model reduction.

The experiments were not done in trplicates, so we can't trust their quality and the authors can't make comparison tests between the results of the experiments

Response: Triplicate experiments were performed in all the assays performed for the optimized extracts. With regards to the response surface methodology, as a Box-Behnken Design with one replication and three center points, triplicate experiments were performed in the center point both for total phenolic and for DPPH evaluation.

The mathematical models (Tables 3) are not correct.

Response: Mathematical model of DPPH was corrected.

Reviewer 3 Report

The manuscript (moleculas-2129644) studied ultrasound technology to extract phenolic compounds from Oregano, and evaluated the antioxidant activity of extraction. After carefully read the response letter and the revised manuscript, I’m afraid the work still need further improvement before publication. Below are some major comments for the authors' consideration:

1.      The title need to revise. Since HPLC-PDA and Q-TOF-MS/MS are identification methods for extracted compounds, they are not the extraction method or antioxidant activity evaluated method.

2.      The abstract need to improve. The authors should reorganize as following order: (Problem of research, aim of study, remarkable methodology, remarkable results, and significance of study). Since the aim of this work was to extract the phenolic compounds from Oregano, it is better to present the results of response surface methodology.

3.      The necessity and innovation of this study are not clear. I guess to mention them in the abstract and introduction.

4.      The figures need to improve! I have suggested before, but the authors didn’t change anything. The figures should be rebuild by Originlab or other scientific mapping software. It should be readability for readers at least!

5.      The authors should mention the added quantitative data (Table 2) in the manuscript.

6.      The authors didn’t explain the reason why the polyphenol content (362.1 ± 1.8 mg GAE/g DM) tested in this manuscript were significantly higher than the previous studies (https://doi.org/10.1016/j.fbp.2020.07.017).

7.      The conclusions need to be re-organized. The authors should highlight the key results of this study.

8.      In line 59, “prevent the oxidating stress”, please support this with references.

9.      Sentence in line 164-166 need improve.

Author Response

Reviewer 3:

  1. The title need to revise. Since HPLC-PDA and Q-TOF-MS/MS are identification methods for extracted compounds, they are not the extraction method or antioxidant activity evaluated method.

Response: The title was revised.

  1. The abstract need to improve. The authors should reorganize as following order: (Problem of research, aim of study, remarkable methodology, remarkable results, and significance of study). Since the aim of this work was to extract the phenolic compounds from Oregano, it is better to present the results of response surface methodology.

Response: Details were incorporated.

  1. The necessity and innovation of this study are not clear. I guess to mention them in the abstract and introduction.

Response: Details about the necessity and innovation of the study were incorporated in lines 90-98 of the revised manuscript.

  1. The figures need to improve! I have suggested before, but the authors didn’t change anything. The figures should be rebuild by Originlab or other scientific mapping software. It should be readability for readers at least!

Response: The quality of the graphs was improved by applying the highest quality that minitab software can offer (from 300 dpi to 1000 dpi). Kindly notice that the minitab is the only software that we can currently use.

  1. The authors should mention the added quantitative data (Table 2) in the manuscript.

Response: Data were incorporated in the manuscript (line 158).

  1. The authors didn’t explain the reason why the polyphenol content (362.1 ± 1.8 mg GAE/g DM) tested in this manuscript were significantly higher than the previous studies (https://doi.org/10.1016/j.fbp.2020.07.017).

Response:  Explanation was incorporated in line 183 in the revised manuscript.

  1. The conclusions need to be re-organized. The authors should highlight the key results of this study.

Response:  Conclusion were re-organised.

  1. In line 59, “prevent the oxidating stress”, please support this with references.

Response: References were incorporated.

  1. Sentence in line 164-166 need improve.

Response: The sentence was improved (Line 177 in the revised manuscript).

Round 3

Reviewer 2 Report

I think that the authors did not need to do a Box-Cox transformation in the treatment since the chosen quadratic model was validated from the beginning. however the treatment with Box-Cox transformation can also be adopted.

The quality of contour plot is still poor.

Author Response

Reviewer 2:

I think that the authors did not need to do a Box-Cox transformation in the treatment since the chosen quadratic model was validated from the beginning. however the treatment with Box-Cox transformation can also be adopted.

Response: We thank the reviewer for his advice.

The quality of the contour plot is still poor.

Response: We thank the reviewer for this comment. The graphs were prepared from scratch as tif files with 1000 dpi resolution.

Reviewer 3 Report

The manuscript (moleculas-2129644) studied ultrasound technology to extract phenolic compounds from Oregano, and evaluated the antioxidant activity of extraction. After carefully read the response letter and the revised manuscript, I consider this work can be accepted. Below are some suggestions for the authors' consideration:

1.      Line 21, “higher activity” can use “multiple bioactivities” instead.

2.      Line 98-100 need to improve.

3.      Line 102-104 need revise.

Author Response

Reviewer 3:

  1. Line 21, “higher activity” can use “multiple bioactivities” instead.

      Response: The sentence was changed according to the reviewer’s advice.

  1. Line 98-100 need to improve.

      Response: The sentence was improved according to the reviewer’s comment.

  1. Line 102-104 need revise.

Response: The text was revised according to Reviewer’s advice